# 3-Methylglutaconic Aciduria Type I Due to *AUH* Defect: The Case Report of a Diagnostic Odyssey and a Review of the Literature

**DOI:** 10.3390/ijms23084422

**Published:** 2022-04-16

**Authors:** Francesca Nardecchia, Anna Caciotti, Teresa Giovanniello, Sabrina De Leo, Lorenzo Ferri, Serena Galosi, Silvia Santagata, Barbara Torres, Laura Bernardini, Claudia Carducci, Amelia Morrone, Vincenzo Leuzzi

**Affiliations:** 1Department of Human Neuroscience, Sapienza University of Rome, 00185 Rome, Italy; serena.galosi@uniroma1.it (S.G.); vincenzo.leuzzi@uniroma1.it (V.L.); 2Laboratory of Molecular Biology of Neurometabolic Diseases, Neuroscience Department, Meyer Children’s Hospital, 50139 Florence, Italy; anna.caciotti@meyer.it (A.C.); lorenzo.ferri@meyer.it (L.F.); amelia.morrone@meyer.it (A.M.); 3Department of Experimental Medicine, Sapienza University of Rome, 00161 Rome, Italy; t.giovanniello@policlinicoumberto1.it (T.G.); s.santagata@policlinicoumberto1.it (S.S.); claudia.carducci@uniroma1.it (C.C.); 4Department of Translational and Precision Medicine, Sapienza University of Rome, 00161 Rome, Italy; s.deleo@policlinicoumberto1.it; 5Medical Genetics Division, IRCCS Casa Sollievo della Sofferenza Foundation, 71013 San Giovanni Rotondo, Italy; b.torres@css-mendel.it (B.T.); l.bernardini@css-mendel.it (L.B.); 6Department of Neurosciences, Psychology, Pharmacology and Child Health, University of Florence, 50134 Florence, Italy

**Keywords:** 3-methylglutaconyl-coenzyme A hydratase (MGH) deficiency, 3-methylglutaconic aciduria type I, *AUH* defect, 3-OH-isovalerylcarnitine, newborn screening, case report

## Abstract

3-Methylglutaconic aciduria type I (MGCA1) is an inborn error of the leucine degradation pathway caused by pathogenic variants in the *AUH* gene, which encodes 3-methylglutaconyl-coenzyme A hydratase (MGH). To date, MGCA1 has been diagnosed in 19 subjects and has been associated with a variable clinical picture, ranging from no symptoms to severe encephalopathy with basal ganglia involvement. We report the case of a 31-month-old female child referred to our center after the detection of increased 3-hydroxyisovalerylcarnitine levels at newborn screening, which were associated with increased urinary excretion of 3-methylglutaconic acid, 3-hydroxyisovaleric acid, and 3-methylglutaric acid. A next-generation sequencing (NGS) panel for 3-methylglutaconic aciduria failed to establish a definitive diagnosis. To further investigate the strong biochemical indication, we measured MGH activity, which was markedly decreased. Finally, single nucleotide polymorphism array analysis disclosed the presence of two microdeletions in compound heterozygosity encompassing the *AUH* gene, which confirmed the diagnosis. The patient was then supplemented with levocarnitine and protein intake was slowly decreased. At the last examination, the patient showed mild clumsiness and an expressive language disorder. This case exemplifies the importance of the biochemical phenotype in the differential diagnosis of metabolic diseases and the importance of collaboration between clinicians, biochemists, and geneticists for an accurate diagnosis.

## 1. Introduction

3-Methylglutaconyl-coenzyme A hydratase (MGH, EC 4.2.1.18) deficiency, or 3-methylglutaconic aciduria type I (MGCA1, MIM ID #250950), is an ultrarare inborn error of the leucine degradation pathway caused by pathogenic variants in the *AUH* gene [1,2,3].

Leucine catabolism consists of a six-step process that mainly takes place in the mitochondria (Figure 1). The first two steps are shared with the other branched-chain amino acids, isoleucine and valine. In humans, any one of these steps (and its corresponding gene) can be impaired, causing a characteristic disease. The most-known diseases associated with the alteration of this catabolic pathway are Maple Syrup urine disease (MSUD), caused by a defect in the branched-chain α-keto acid dehydrogenase complex (BCKDH), and Isovaleric Acidemia (IVA), caused by a defect in the isovaleryl CoA dehydrogenase gene (*IVD*). Different phenotypes, encompassing severe neonatal-onset forms with metabolic decompensation, are recognized, and their management and treatment are well established [4]. A more-benign disease of the leucine degradation pathway is also known, 3-Methylcrotonyl-CoA carboxylase deficiency, due to *MCCC1* and *MCCC2* gene defects. Phenotypes have been recognized thanks to the implementation of newborn screening that has allowed for the identification of asymptomatic newborn infants, siblings, and mothers [4].

MGH catalyses the fifth step in the leucine degradation pathway, the reversible hydration of 3-methylglutaconyl-coenzyme A (3-MG-CoA) to 3-hydroxy-3-methylglutaryl-coenzyme A (HMG-CoA). MGH deficiency results in 3-MG-CoA accumulation within the mitochondrial matrix [5,6], which in turn is converted to 3-methylglutaconic acid by coenzyme A thioester bond hydrolytic cleavage, followed by export of 3-methylglutaconic acid from the mitochondrion [7].

MGCA1 has been diagnosed in 19 subjects described in the literature (Table 1 and Table 2). Patients affected by this disease manifest a wide range of clinical signs, ranging from no or mild symptoms [8,9] to mild neurological impairment [10,11], acute encephalopathy [12,13,14], severe encephalopathy with basal ganglia involvement [15,16], and slowly progressive leukoencephalopathy presenting in adulthood [17,18,19,20]. Dilated cardiomyopathy [21], central precocious puberty, attention deficient hyperactivity disorder, learning disability, and white matter alterations on magnetic resonance imaging [22] have also been described.

Here, we report the successful diagnostic odyssey of a neonate positive on newborn screening for 3-methylglutaconic aciduria who was diagnosed with MGCA1 through clinical–biochemical–genetic integration.

## 2. Results

### 2.1. Case Report

The 31-month-old girl here reported was referred to our center after the detection of increased levels of 3-hydroxyisovalerylcarnitine (C5OH) at newborn screening, which were confirmed in subsequent samples (C5OH between 0.8 and 1.1 µmol/L; cut-off for newborn screening 0.59 at first test and 0.55 afterwards). Biotinidase activity was within the normal range, while urinary organic acid analysis revealed high excretion of 3-methylglutaconic acid associated with a moderate increase in 3-hydroxyvaleric acid and a slight increase in glutaric acid and 3-methylglutaric acid. Her mother’s acylcarnitine profile and urinary organic acid analysis were normal, suggesting 3-methylglutaconic aciduria in the patient. The patient was the only child of a non-consanguineous healthy couple. During the pregnancy, her mother experienced symptomatic lumbar disc herniation requiring anti-inflammatory and steroidal therapy and a planned caesarean delivery at term (38 weeks + 2 days). The child’s parameters at birth were normal (weight 2.9 kg, 11th percentile; length 49 cm, 29th percentile; head circumference 33 cm, 14th percentile). At the first admission to our center, she showed a normal general and neurological examination, and routine blood test results and heart ultrasound were normal. 

### 2.2. Diagnostic Procedures

Due to the wide differential diagnosis in 3-methylglutaconic acidurias, the patient underwent a next-generation sequencing (NGS) panel containing 16 causative genes of 3-methylglutaconic acidurias. The procedure did not identify any genetic alterations, either in coding regions or in intron/exon boundaries, with the exception of a heterozygous pathogenetic variant in the ryanodine receptor 1 (RYR1) gene (MIM ID* 180901) (c.14126C > T; p.Thr4709Met [23]) inherited from her healthy mother. This variant was previously found in a heterozygous state in a child with congenital myopathy and the authors demonstrated that this particular pathogenic variant had a monoallelic expression in muscle when inherited from the father [24]. Our patient did not show signs of myopathy and her mother, who carried the same pathogenic variant, did not show signs of myopathy or 3-methylglutaconic aciduria.

To further investigate the strong biochemical indication, we measured MGH activity. The enzyme assay revealed markedly reduced MGH activity in the lymphocytes of our patient (<0.02 nmol/(min.mg prot), reference values 1.4–4.8 nmol/(min.mg prot)), confirming the biochemical diagnosis of MGCA1. 

Single nucleotide polymorphism (SNP) array analysis was then performed in order to verify the presence of AUH intragenic microdeletions. This analysis disclosed two distinct microdeletions of 45 kb and 16 kb encompassing this gene; the former was covered by 43 markers, mapped to chr9:93936592_93981126 (hg19), and partially included the RefSeq gene LINC00484 and exons 8–10 of AUH (NM_001698.3), while the latter was covered by 76 markers, mapped to chr9:94106723_94122848 (hg19), and included exons 2 and 3 of the gene (Figure 2). 

The same analysis also disclosed a maternally inherited microduplication at 1p36.33 that included the following genes: FNDC10, LOC105378586, ATAD3C, ATAD3B, SSU72, ATAD3A, and TMEM240, though its clinical significance is currently unknown. Copy number variation (CNV) analysis performed on patient and parental samples confirmed the presence of the two deletions in compound heterozygosity, with the 45 Kb deletion present in the maternal sample and the 16 kb deletion inherited from the father (Figure 3).

The patient’s deletions include exons 2–3 and exons 8–10. The patient shares the deletion of exons 2–3 with her father and the deletion of exons 8 to 10 with her mother.

### 2.3. Clinical Follow-Up

During the follow-up, C5OH tended to increase up to 3.33 µmol/L (r.v. 0.08–0.44), while organic acid excretion remained stable over time. 

As soon as the enzyme activity assay result arrived (at 20 months old), the patient started levocarnitine at 80 mg/kg/day and her dietary protein intake was slowly decreased.

At the last examination (31 months of age), her statural and ponderal growth were normal (height 26th percentile, weight 74th percentile, head circumference 52nd percentile), and she showed mild clumsiness and an expressive language disorder.

## 3. Discussion

MGH deficiency was first hypothesized on a biochemical basis due to the excessive excretion of 3-methylglutaconic acid (3-MGA), 3-methylglutaric acid, and 3-hydroxyisovaleric acid in two siblings suffering from a language disorder [5]. Reduced MGH activity was then demonstrated by Narisawa and colleagues [6] in patient fibroblasts, which identified the last defect in the leucine catabolic pathway. Approximately 15 years later, the molecular cause of the disease was independently identified in the *AUH* (*600529) gene by two research groups [1,3]. Many of the gene variants described thus far in compound heterozygosity or homozygosity are predicted to lead to complete loss of protein function [17].

While 3-MGA is found only in traces in healthy individuals, an increased level of this branched-chain organic acid is a relatively common finding in patients suspected of having a metabolic disorder (about 3% in [25]), such as organic acidurias, glycogen storage disorders, fatty acid oxidation disorders, and urea cycle disorders, usually during metabolic decompensation and in association with more disease-specific urinary metabolites [25]. This increase is most likely due to mitochondrial dysfunction, and indeed patients with mitochondrial depletion syndromes and mitochondrial disorders in general (e.g., POLG, SUCLG1, SUCLA2, and TWINKLE) often show 3-methylglutaconic aciduria [26]. However, in some patients increased 3-MGA excretion is a major finding, being the hallmark of the disease and often paramount in the diagnosis. Therefore, the “inborn errors of metabolism with 3-methylglutaconic aciduria as a discriminative feature” label has been proposed for this group of conditions, which consists of two categories: primary 3-methylglutaconic acidurias and secondary 3-methylglutaconic acidurias [25]. Primary 3-methylglutaconic acidurias are disorders of mitochondrial HMG-CoA metabolism and are caused by MGH (*AUH*) deficiencies, though some authors have also suggested a role of HMG-CoA lyase (*HMGCL*) defects, an enzyme common to ketogenesis and leucine oxidation pathways [27,28,29]. In the urine of patients with primary 3-methylglutaconic acidurias, there is also excess 3-hydroxyisovaleric acid derived from unmetabolized 3-hydroxyisovaleryl-coenzyme A ester, a leucine oxidation intermediate from just upstream of 3-MG-CoA, and subsequent HMG-CoA formations.

Secondary 3-methylglutaconic acidurias include disorders of phospholipid remodelling (such as in TAZ defects/Barth syndrome, SERAC1 defects/MEGDEL syndrome, and Sengers syndrome), mitochondrial membrane-associated proteins (such as in OPA3 defects/Costeff syndrome, DNAJC19 defects/DCMA syndrome, and TMEM70 defects), or unknown origin (as in CLPB defects and in 3-methylglutaconic acidurias not otherwise specified), and the list of disease-causing genes is growing [4,30,31,32].

The natural history of MGCA1 is not known and patients diagnosed with this disease may have variable clinical symptoms (Table 2). The percentage of consanguinity in the parents of affected patients is quite high (Table 1) and some authors question whether the severe clinical picture of some patients is related to the MGH defect or to another, still undiscovered, genetic defect.

Tandem mass spectrometry now allows newborn screening for more than 40 metabolic disorders simultaneously, but evidence of its effectiveness for some diseases has yet to be established due to the low prevalence and lack of a clinical phenotype. For these reasons, diseases amenable to newborn screening are often listed in primary/core and secondary panels according to whether evidence about the effectiveness of newborn screening and treatment on disease outcome is high or low, respectively [33]. 

The first patients found to have MGCA1 were diagnosed before the introduction of newborn screening and had a variable phenotype. The introduction of newborn screening allowed for the detection of MGCA1 based on a biochemical phenotype in asymptomatic newborns. MGCA1 belongs to the secondary panel of newborn screening programs due to the low number of patients described and the uncertainty about the clinical phenotype. However, it shares primary metabolic markers with other diseases belonging to the primary panel (e.g., ß-ketothiolase deficiency), which allows for its early diagnosis but leads to questions about follow-up and dietetic treatment indications.

The patient described in this paper is a healthy child that was put on levocarnitine supplementation and started a slow dietary restriction of protein intake after the MGCA1 diagnosis was made as a preventive measure for possible later alterations. At the last examination, when she was 31 months old, she showed mild clumsiness and an expressive language disorder, which is a quite common finding at this age, also in “metabolically healthy” children. Further follow-up is needed to assess the outcome of this condition in this patient diagnosed by means of newborn screening.

This case exemplifies the importance of the biochemical phenotype in guiding the differential diagnosis, at least in metabolic diseases with a biochemical marker. In this case, only the analysis of the biochemical alteration pattern allowed for further investigation after a negative NGS panel. Fortunately, the enzyme activity assay confirmed the suspicion, and SNP array and CNV analyses using real-time polymerase chain reaction (PCR) assays subsequently detected two different deletions in compound heterozygosity, a rare finding in an ultrarare disease, ultimately confirming MGCA1. Thus far, only one intragenic deletion involving exons 1–3 has been reported in a homozygous state in a patient affected by MGCA1 [8], and only two cases involving exon 8 and exons 2–4 are reported in international databases (https://www.ncbi.nlm.nih.gov/clinvar/; accessed on 24 February 2022). The confirmed diagnosis in our patient also highlights the importance of collaboration between clinicians, biochemists, and geneticists.

Finally, this case also describes the challenges of managing patients and parents of children who are diagnosed early as a result of newborn screening. In this case, the challenges included the course of action and communication with the patient’s parents and the decision to start dietetic treatment, which needed to be balanced according to the risk of developing symptoms and the burden of dietetic treatment in the absence of sufficient evidence to guide this decision.

## 4. Materials and Methods

### 4.1. Case Report

This paper was written in accordance with the CAse REport (CARE) guidelines (https://www.care-statement.org, accessed on 10 April 2022).

### 4.2. NGS

Genomic DNA of the patient and her parents was extracted from peripheral blood using a QIAsymphony instrument (Qiagen, Hilden, Germany). We performed targeted NGS resequencing of 16 genes associated with 3-methylglutaconic aciduria using a custom-designed panel (Illumina, San Diego, CA, USA). We prepared libraries using the Nextera Rapid Capture Enrichment kit (Illumina, San Diego, CA, USA) according to the manual instructions. Libraries were sequenced by a paired-end 2 × 150 bp protocol on a MiSeq System (Illumina, San Diego, CA, USA) to obtain an average coverage of above 100x, with >95% of target bases covered at least 15x. For data analysis, we used the BWA, Picard, and GATK tools. Variant annotation was performed by the ANNOVAR tool. The reference sequences used for the nomenclature of exons and variants in the AUH and RYR1 genes were NM_001698.3 and NM_000540.3, respectively.

### 4.3. SNP Array Analysis

CNV analysis was carried out using an SNP-array platform (Cytoscan HD, Thermo Fisher Scientific, Inc., Waltham, MA, USA) following the manufacturer’s recommendations. Briefly, 250 ng of genomic DNA extracted from lymphocytes (NucleoSpin Blood, Macherey-Nagel, Duren, Germany) was digested with NspI. After digestion, an adaptor was linked to the restricted fragments that were then amplified by PCR. After purification using magnetic beads, PCR products were fragmented and end-labeled using a terminal deoxynucleotidyl transferase, and then hybridized for 16–18 h to the Cytoscan HD chip at 50 °C in a GeneChip Hybridization Oven 640 (Thermo Fisher Scientific, Inc., Waltham, MA, USA). The chips were washed, stained in a GeneChip Fluidics Station 450, and scanned with a GeneChip 3000 7G scanner. Data were analyzed with ChAS software (v4.2.1; Thermo Fisher Scientific, Inc., Waltham, MA, USA). Two hundred and seventy healthy controls belonging to the International HapMap Project were used as a reference sample in data analysis (Thermo Fisher Scientific, Inc., Waltham, MA, USA). Genome-wide microdeletion/microduplication (CNV) analysis was performed only calling losses and gains covered by at least 25 markers and with a minimum length of 75 Kb, and CNVs were classified according to the American College of Medical Genetics and Genomics (ACMG) and ClinGen [34]. CNVs within the candidate gene were analyzed without a filter.

### 4.4. CNV Assays

*AUH* CNV assays were performed combining TaqMan^®^ MGB probe chemistry with real-time PCR instruments (Applied Biosystems^®^ 7500 Real-Time PCR) (Thermofisher Italia, Monza, Italy). The methods employed were as previously described [35].

Human Predesigned TaqMan^®^ Copy Number Assays (FAM dye-labeled) to quantify the *AUH* gene (exons 2, 3, 7, 8, and 10 cat. Hs02688800; Hs02216634; Hs02965517; Hs02900505; Hs02517722) were run in triplicate with human RNaseP TaqMan^®^ Copy Number Reference Assays (VIC dye-labeled, cat. 4403326) in duplex real-time PCR reactions. The reaction conditions were the following: 10 ng gDNA and 1×TaqMan probe/primer mix in 1×TaqMan Universal Master Mix in a 25-µL reaction. The cycling conditions were 20 s at 50 °C and 10 min at 95 °C, followed by 40 cycles of 15 s at 95 °C and 60 s at 60 °C.

TaqMan^®^ Copy Number Assays quantify *AUH* exons normalized on the endogenous reference RNaseP gene. Normalizations were performed by relative quantifications determined by the ΔΔCt ((FAM Ct- VIC Ct) sample—(FAM Ct—VIC Ct) calibrator) method as previously reported [36]. The gene copy number is two times the relative quantity [36].

## Figures and Tables

**Figure 1 ijms-23-04422-f001:**
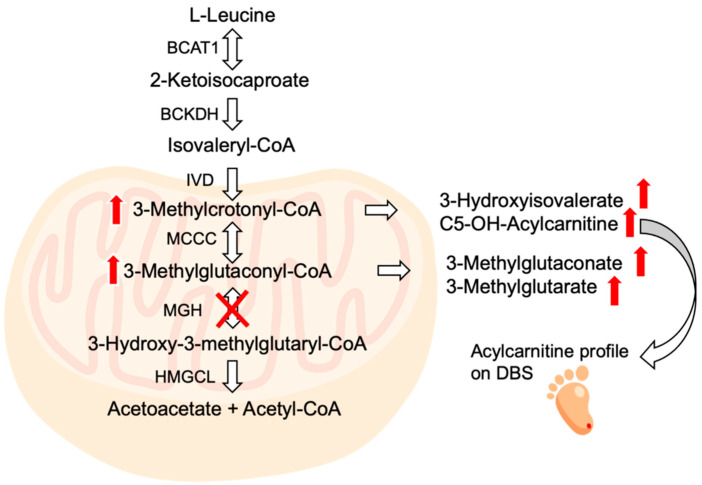
Illustration of the leucine catabolic pathway. BCAT1, branched-chain amino acid aminotransferase 1; BCKDH, branched-chain α-keto acid dehydrogenase complex; IVD, isovaleryl-CoA dehydrogenase; MCCC, 3-methylcrotonyl-CoA carboxylase; 3-MGH, 3-methylglutaconyl-CoA hydratase; HMGCL, 3-hydroxy-3-methylglutaryl-CoA lyase; 3-HIVA, 3-hydroxyisovaleric acid; DBS, dried blood spot [4].

**Figure 2 ijms-23-04422-f002:**
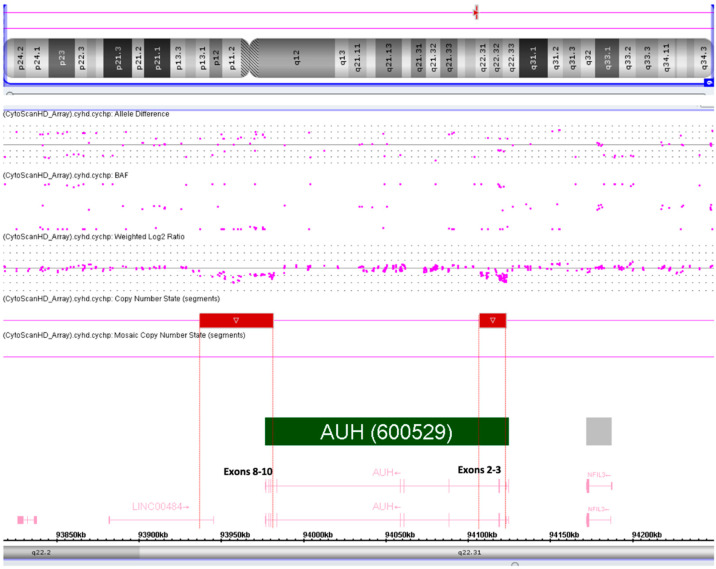
Single nucleotide polymorphism array profile of the 9q22.31 region showing the two losses (red bars and dotted red lines) involving exons 8–10 and 2–3 of the AUH gene in the 9q22.31 region.

**Figure 3 ijms-23-04422-f003:**
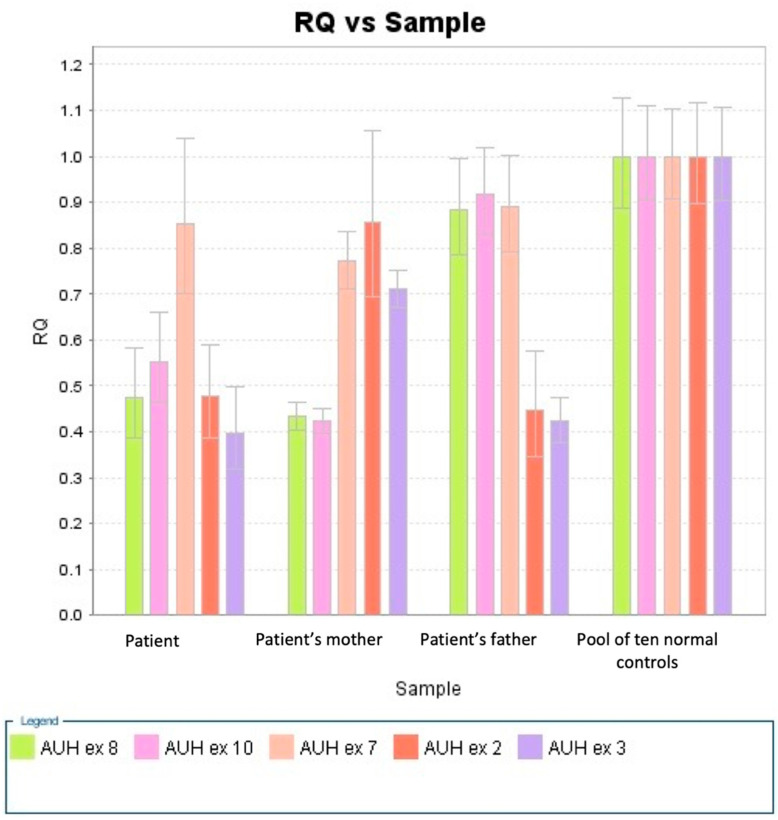
Relative quantification (RQ) values of AUH exons by sample.

**Table 1 ijms-23-04422-t001:** Genetic, biochemical, and instrumental data on all *AUH*-deficient patients reported in the literature.

References	Ancestry (Consanguinity)	*AUH* Variant	MGH Activity (r.v.)	Urinary 3-MGA (r.v.)	Urinary 3-HIVA (r.v.)	Neuroimaging/ Age at	MRI: WM Changes	MRI: Additional Findings
ID 1 [3,5,6,20]	Moroccan (-)	c.589C > T p.(Arg197*)/c.589C > T p.(Arg197*)	F: 11 ± 4 pmol/min per mg protein (495 ± 89)	519–840 mmol/mol creat (<5.66)	144–206 mmol/mol creat (<8.15)	NA	NA	NA
ID 2 [1,3,5,6,20]	Moroccan (-)	c.589C > T p.(Arg197*)/c.589C > T p.(Arg197*)	F: < 0.1 nmol/min/mg protein (2.8 ± 0.6)	762–930 mmol/mol creat (<5.66)	172–264 mmol/mol creat (<8.15)	NA	NA	NA
ID 3 [12]	NA (NA)	NA	F: 3-HB/CoA esters 0.5 (2–63) L: 3-HB/CoA esters 1–3 (7–20) *	Marked elevation	Marked elevation	CT normal	NA	NA
ID 4 [3,10,13,20]	NA (NA)	c.719C > T p.(Ala240Val)/c.613dupA p.(Met205Asnfs*5)	F: 3-HB/CoA esters 0.2–0.55 (5.3 ± 1.2) *	249–953 mmol/mol creat (<9)	47–443 mmol/mol creat (<46)	CT normal	NA	NA
ID 5 [10]	White (NA)	NA	F: 3-HB/CoA esters 0.16–0.35 (5.3 ± 1.2) *	Marked elevation	Marked elevation	NA	NA	NA
ID 6 [10]	NA (+)	NA	F: 3-HB/CoA esters 0.16–0.58 (5.3 ± 1.2) *	482–1153 mmol/mol creat (<9)	374–3840 mmol/mol creat (<46)	MRI	Abnormal signals in the white matter	Globus pallidum hyperintensity images, cavum septum pellucidum, and cavum vergae
ID 7 [15,16,20]	Japanese (+)	c.263-2A > G/c.263-2A > G	F: 3-HB/CoA esters 0.3 (5.0–10.6) * L: 3-HB/CoA esters 0.2 (2.3–5.7) *	168 mmol/mol creat (0–15)	292 mmol/mol creat (0–4)	MRI/9 months and 23 months	Myelination almost normal	Cerebral atrophy and progressive basal ganglia atrophy
ID 8 [1,3,11,20]	Afghan (+)	c.895-1G > A/c.895-1G > A	F: < 0.1 nmol/min/mg protein (2.8 ± 0.6)	400 mmol/mol creat (<9)	Increased	MRI/paediatric	NR	NR
ID 9 [3,20]	Lebanese (+)	c.80delG p.(Ser27Metfs*8)/c.80delG p.(Ser27Metfs*8)	F: 3-HB/CoA esters 0.06 (1.2–4.5) *	1500–2900 mmol/mol creat	No increase	NA	NA	NA
ID 10 [17,20]	German (-)	c.943-2A > G/c.943-2A > G	F: markedly decreased	570 mmol/mol creat (<9)	500 mmol/mol creat (<67)	MRI/10 years	Mild abnormalities in deep frontal WM sparing the corpus callosum and the U-fibers	NR
ID 11 [18,20]	Dutch (-)	c.559G > A p.(Gly187Ser)/c.650G > A p.(Gly217Asp)	F: undetectable (2.1 ± 0.7 nmol/min/mg protein)	94–141 mmol/mol creat (1.0–6.5) #	61–63 mmol/mol creat (3–15)	MRI/61 years	Confluent lesions of deep and subcortical WM, sparing the corpus callosum and periventricular rim	Cranial MRS: 3-HIVA peak
ID 12 [19,20]	NA (+)	c.895-1G > A/c.895-1G > A	NA	108 mmol/mol creat (<4.2)	NA	MRI/adult	WM hyperintensity extending into the subcortical U-fibers and middle cerebellar peduncle	NR
ID 13 [20]	British (+)	c.991A > T p.(Lys331*)/c.991A > T p.(Lys331*)	F/L: undetectable	78 mmol/mol creat	Mildly elevated	MRI/50 years	Confluent lesions of deep and subcortical WM, sparing the corpus callosum and periventricular rim and parietooccipital regions	NR
ID 14 [8]	Pakistani (+)	Homozygous deletion of exons 1–3	NA	174.24 mmol/mol creat (<12.42)	88.7 mmol/mol creat (<37.7)	MRI	Bilateral patchy hyperintensity in the frontal and parietal subcortical WM	Cranial MRS: 3-HIVA peak
ID 15 [8]	Pakistani (+)	Homozygous deletion of exons 1–3	F: undetectable (7.7 ± 1.4 nmol/min mg)	261.4 mmol/mol creat (<12.42)	242.6 mmol/mol creat (<37.7)	NA	NA	NA
ID 16 [21]	Canadian (NA)	NA	F: reduced	Increased	NA	NA	NA	NA
ID 17 [9]	NA (+)	c.179delG p.(Gly60Valfs*12)/c.179delG p.(Gly60Valfs*12)	NA	Increased	Increased	MRI/2 years and 3.5 years	Delayed myelination in the trigone region, a few nonspecific hyperintensities in the centrum semiovale improved at the second study (treatment?)	NR
ID 18 [14]	NA (+)	c.505 + 1G > C/c.505 + 1G > C	NA	Increased	Increased	MRI/3 years	NR	NR
ID 19 [22]	Caucasian (-)	c.330 + 5G > A/c.330 + 5G > A	L: 0.02 nmol/min/mg protein (1.4–4.6)	Increased	Increased	MRI/6, 9, and 10 years	Progressive hyperintensive lesions in the centrum semiovale, bilateral subcortical frontal WM, and the deep frontoparietal WM	NR
ID 20	Italian (-)		L: <0.02 nmol/min/mg protein (1.4–4.6)	Marked elevation	Marked elevation	NA	NA	NA

* The ratio of 3-hydroxybutyric (3-HB) acid to total 14C-labeled CoA esters (predominantly 3-MG-CoA) was used to assess MGH activity; # measured 1h-NMR. 3-HIVA, 3-hydroxyisovaleric acid; 3-MGA, 3-methylglutaconic acid; CT, computed tomography; MRS, magnetic resonance spectroscopy; NA, not available; NR, not reported; MRI, magnetic resonance imaging; r.v., reference values; WM, white matter.

**Table 2 ijms-23-04422-t002:** Clinical data on all *AUH*-deficient patients reported in the literature.

References	Age at Onset/ Gender	Age at Last Examination	Decompensation/ Acute Encephalopathy	Neurodevelopmental Disorders	Neurological Phenotype	Other Clinical Features	Treatment		
							Levocarnitine (Dosage)	Leucine/Protein Restricted Diet	Outcome after/Effect of Treatment
ID 1 [3,5,6,20]	7 years/M—sib	7 years	One attack of unconsciousness that lasted for almost a day. Fasting hypoglycaemia	GDD	Normal	Nocturnal enuresis	NR	NR	NR
ID 2 [1,3,5,6,20]	-/M—sib	5.4 years	NR	Speech disorder	Normal	NR	NR	NR	NR
ID 3 [12]	4 months/M	NA	One episode of decompensation during viral illness	NR	Normal	Bronchiolitis and gastro-oesophageal reflux	Yes (unknown)	Leucine intake 100–120 mg/kg/day	Normalization of metabolic alterations
ID 4 [3,10,13,20]	Neonatal/F	13 years	Vomiting after birth. Relapsing encephalopathy during upper respiratory infections	NR	Insomnia, irritability after birth, persistent crying fits, self-mutilation	Hepatomegaly	Yes (unknown)	Yes	Ineffective leucine-restricted diet. L-carnitine: improvement of clinical (no feeding difficulties and insomnia, hepatomegaly cleared) and metabolic alterations
ID 5 [10]	Neonatal/M	NA	Severe acidosis, foetal distress with hypoxic ischaemic encephalopathy	Moderate GDD	Severe dystonic cerebral palsy with generalized hyperreflexia and irritability	Gastro-oesophageal reflux	Yes (unknown)	Low-protein diet using natural foods (1 g/kg/day)	No more episodes of vomiting, lethargy, and acidosis nor developmental plateau or regression
ID 6 [10]	-/M	4 years	Seizures possibly associated with episodes of hypoglycaemia	Severe GDD	Severe hypotonia and intermittent thrusting in the sitting position, seizures	Premature birth, frequent upper respiratory infections, lumbar scoliosis, hepatomegaly	Yes (unknown)	NR	NA
ID 7 [15,16,20]	4 months/M	43 months	Persistent metabolic acidosis	Profound GDD	Spastic quadriplegia, athetoid dystonic movements of upper limbs	Failure to thrive	NR	Protein-restricted diet (1.0 g/kg per day; L-leucine 80 mg/kg per day) using a leucine-free formula	Improvement of metabolic alteration, marginal clinical improvement (reacquisition of eye-pursuits and social smiles)
ID 8 [1,3,11,20]	1 year/M	4 years	NR	GDD 4y: IQ 83, sustained attention deficit	Normal	Primary nocturnal and diurnal enuresis	NR	Restricted protein intake (1.5 g/kg body weight per day)	Clinical improvement
ID 9 [3,20]	NBS/M—sib	9 years	NR	NR	Normal	NR	NR	NR	NR
ID 10 [17,20]	1 year/M	10 years	NR	Attention-deficit/hyperactivity disorder	Febrile seizures (15 up to age 7)	NR	NR	NR	Mild improvement of seizure severity with emergency protocol
ID 11 [18,20]	35 years/F	61 years	NR	NR	Progressive spastic ataxia	NR	NR	NR	NR
ID 12 [19,20]	52 years/F	55 years	NR	NR	Dementia, spasticity, ataxia	Urinary incontinence at age 52	NR	NR	NR
ID 13 [20]	30 years/M	59 years	NR	NR	Progressive spastic ataxia and dementia	NR	NR	NR	NR
ID 14 [8]	10 years/F—sib	14 years	NR	IQ 82 at 10 years, arithmetic and learning disability, attention deficit	Mild uncoordination	NR	50 mg/kg/day	Modified protein intake (1 g/kg/day natural protein intake)	Stable course
ID 15 [8]	3 years/M—sib	9 years	NR	Severe expressive language delay at 3 years	Dysarthria, two febrile seizures	History of dislike of meat	50 mg/kg/day	Modified protein intake (1 g/kg/day natural protein intake)	Stable course
ID 16 [21]	1 year/M	25 years	Encephalopathy with metabolic decompensation during sepsis	Motor deficit, developmental learning delays	Cyanotic breath-holding spells, seizures at 1 year	Renal and heart failure during sepsis at 25 y, dilated cardiomyopathy	NR	NR	NR
ID 17 [9]	22 months/M	3.5 years	NR	Delayed milestone achievements with near normal cognition	Central hypotonia, intention tremor	Moderate sensorineural hearing loss	100 mg/kg/day in three doses	Restricted leucine diet (60 mg/kg/day)	Incoordination, neurocognitive improvement
ID 18 [14]	3 years/F	5 years	Status dystonicus during febrile illness, irritability, unable to sleep	NR	Normal	NR	NR	Low-leucine diet	No further dystonic relapses after two years
ID 19 [22]	4.5 years/F	11 years	NR	Learning disability, attention deficit	Central hypotonia, intention tremor, and dysdiadochokinesia	Central precocious puberty	85 mg/kg/day	60 mg/kg/day	Improvement of attention
ID 20	NBS/F	31 months	NR	Expressive language disorder	Clumsiness	NR	80 mg/kg/day	Slow decrease in protein intake	NA

GDD, global developmental delay; F, female; M, male; NA, not available; NBS, newborn screening; NR, not reported.

## Data Availability

Data are available on request. Data are contained within the article.

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
