# Peer review of "3-Methylglutaconic Aciduria Type I Due to AUH Defect: The Case Report of a Diagnostic Odyssey and a Review of the Literature"

_ijms, 2022, doi:10.3390/ijms23084422_

Round 1
Reviewer 1 Report
This is an important case report because it presents a new pathogenic variant in AUH and more importantly highlights the culprit of exome analysis as the first tier diagnostic tool of IEM. The manuscript is includes review of the known cases. Therefore consider add ".......and review of the literature" to the title. The manuscript is well written and I have only two minor questions/comments re the results. 1-could C5OH to C8 acylcarnine ratio provide additional info and a clearer diagnosis ? 2-could the microdeletions not be detected by checking the read -depth of the exome analysis?
Author Response
This is an important case report because it presents a new pathogenic variant in AUH and more importantly highlights the culprit of exome analysis as the first tier diagnostic tool of IEM. The manuscript is includes review of the known cases. Therefore consider add ".......and review of the literature" to the title. The manuscript is well written and I have only two minor questions/comments re the results.
Answer: Thanks for the nice comments, the title has been updated accordingly.
1-could C5OH to C8 acylcarnine ratio provide additional info and a clearer diagnosis ?
Answer: C5OH/C8 ratio is a good marker for the identification of all the conditions associated to C5OH increase and in the first sample of the patient was 31.7 (cut-off 7.1). However, it seems not to be specific for MGCA1 and it does not increase the sensitivity and specificity of test alone. Collaborative Laboratory Integrated Reports (CLIR; https:// clir.mayo.edu) analysis is performed in the evaluation of newborn screening data and in CLIR algorithm C5OH/C8 is included together other 12 AC ratios.
2-could the microdeletions not be detected by checking the read -depth of the exome analysis?
Answer: At first, no deletions in the AUH gene of our patient had been suggested by the analysis of NGS sequencing data with the specific software tool CoNVaDING (Copy Number Variation Detection In Next-generation sequencing Gene panels) by Johansson et al. 2016, Hum Mutat 37:457-464.
After we became aware of the SNP array data, we retrospectively visually checked the NGS reads alignment by IGV software (https://igv.org/) and we could actually visualize a lower read depth in AUH exons 2-3 and AUH exons 8-10 in patient’s DNA with respect to our control DNAs. Read depth was also reduced in AUH exons 2-3 in the DNA from patient’s father, while a reduced read depth was perceptible in the AUH exons 8-10 in the DNA from the patient’s mother. Such analyses, combined with the SNP array data, guided us to design the probes for real-time PCR.
Reviewer 2 Report
This is an interesting case report that shows how difficult it can be to diagnose rare inherited metabolic diseases. Overall, in my view, the main weaknesses are in the structure of the paper.
The title should include the phrase „A case report“.
Please check the manuscript according to CARE (case report guidelines) or an other guideline for case report and add the info oft he guideline use in the methods.
In my opinion the paper should be divided in more and clearer sections (like "case description", NGS", "clinical parameters" and so on). The Methods should be point 2 and not 4, and some parts of the results are more a part of the methods.
Please add in the introduction a short info of the more popular metabolic diseases of this metabolic pathway (BCKD - MSUD; IVD - IVA).
The tables are very interesting, but too long for the main text. Please move table 1 and 2 to the appendix.
In line 65 (p 2) and in the description of figure 1 references are missing.
The lines 149 – 194 are parts for the introduction, and not of the discussion.
The results and/or discussion lacks information about the patient's treatment and outcome.
It should be mentioned that mild clumsiness and language disorders can occur at this age even in metabolically healthy individuals.
Author Response
This is an interesting case report that shows how difficult it can be to diagnose rare inherited metabolic diseases. Overall, in my view, the main weaknesses are in the structure of the paper.
The title should include the phrase „A case report“.
Answer: The title has been rephrased accordingly.
Please check the manuscript according to CARE (case report guidelines) or an other guideline for case report and add the info oft he guideline use in the methods.
Answer: Thanks for this suggestion. The paper now fulfills the CARE checklist: the title and the key words have been updated accordingly. The use of guidelines have been added as methods to the manuscript.
In my opinion the paper should be divided in more and clearer sections (like "case description", NGS", "clinical parameters" and so on).
Answer: The results section has been divided in subsection as requested.
The Methods should be point 2 and not 4, and some parts of the results are more a part of the methods.
Answer: The conventional order of scientific research includes the methods before the results section. However, some journal has started to move it at the end of the paper for an easier reading. Indeed, the IJMS template was used to write the paper and the methods section is required as the point 4. Some part of the results section, such as the NGS analysis, has been moved to the methods section as requested, creating a new subsection.
Please add in the introduction a short info of the more popular metabolic diseases of this metabolic pathway (BCKD - MSUD; IVD - IVA).
Answer: Thanks for this suggestion. In the introduction section a short info of the other diseases has been added.
The tables are very interesting, but too long for the main text. Please move table 1 and 2 to the appendix.
Answer: We agree with the reviewer about the excessive length of the Tables that would better fit the supplementary material. However, as per the other reviewer request, more emphasis on the review of the literature has been given. This way, we think that tables in the main text are important for the review process of a reader.
In line 65 (p 2) and in the description of figure 1 references are missing.
Answer: References has been added as requested. We further updated the figure itself because we noticed a double headed arrow missing in the first-step reaction.
The lines 149 – 194 are parts for the introduction, and not of the discussion.
Answer: We agree with the reviewer. Indeed, those lines were firstly drafted for the introduction section. However, this way the introduction section would have been way too long (also CARE guidelines suggests a short introduction), and it would have anticipated the clinician reasoning that led to further diagnostic steps after a negative NGS panel. We would prefer to leave the historical and biochemical background of MMGCA1 at the beginning of the discussion section.
The results and/or discussion lacks information about the patient's treatment and outcome. It should be mentioned that mild clumsiness and language disorders can occur at this age even in metabolically healthy individuals.
Answer: Thanks for this suggestion, we agree with the reviewer and some info in the discussion section has been added accordingly.